# Acute Effect of Resistant Starch on Food Intake, Appetite and Satiety in Overweight/Obese Males

**DOI:** 10.3390/nu10121993

**Published:** 2018-12-15

**Authors:** Najlaa M. Al-Mana, M. Denise Robertson

**Affiliations:** 1Clinical Nutrition Department, College of Applied Medical Sciences, King Saud bin Abdul-Aziz University for Health Sciences, Jeddah 21423, Saudi Arabia; 2Nutritional Sciences, University of Surrey, Leggett Building, Guildford, Surrey GU2 7WG, UK; m.robertson@surrey.ac.uk

**Keywords:** resistant starch, dietary fibre, glucagon-like peptide-1, appetite, satiety, obesity

## Abstract

Several studies have linked increased intake of dietary fibre to improvement in the management of body weight. Dietary fibre from resistant starch (RS) has been shown to have an impact on food intake in normal weight individuals, but its role in obesity is unknown. The present study aimed to investigate the short-term effects of RS on appetite, satiety and postprandial metabolism in overweight/obese subjects. In this single-blind randomized crossover study, overweight/obese healthy males consumed a test breakfast and lunch containing either 48 g RS or a placebo. Postprandial qualitative appetite, glucose, insulin, and GLP-1 were measured every 30 min for 7 h. Energy intake values from an *ad libitum* dinner and for a 24-h period were assessed. Acute consumption of RS at breakfast/lunch significantly reduced the energy intake at the *ad libitum* dinner (*p* = 0.017). No significant effect over 24 h or qualitative feelings of satiety were observed. Significant treatment × time effects were found for postprandial glucose (*p* = 0.004) for RS compared to placebo, with a trend for higher C-peptide concentrations following RS. The postprandial insulin and GLP-1 responses were not significantly different. RS may indeed have short-term beneficial effects in obese individuals.

## 1. Introduction

The prevalence of obesity and being overweight has been widely reported over the past 20 years [1,2,3,4,5]. The World Health Organization [6] considers the worldwide rise in obesity a global epidemic. Obesity is a problem in not only high-income countries but also low- and middle-income countries, where the prevalence of the condition has increased dramatically, especially in urban areas [7]. However, although obesity is a complex chronic condition, lifestyle modifications that include modifying food behavior may be crucial in obesity management [8].

Several studies have shown an association of increased intake of dietary fibre with improvement in the management of body weight [9,10,11,12,13]. In general, the effect of dietary fibre on weight regulation is manifested through different mechanisms, including stabilising interprandial blood glucose, food intake reductions, delayed gastric emptying time, and intestinal hormone response alterations after fibre consumption [13,14,15].

Resistant starch (RS) is classified as a dietary fibre. RS is indigested in the small intestine and enters the large intestine intact, where it is fermented by gut microbiota to produce short-chain fatty acids [16,17,18]. It has been demonstrated that RS may physiologically act like other types of non-viscous dietary fibres [19,20,21]. Therefore, it could potentially play an important role in weight regulation due to its impact on satiety and food intake.

A limited number of studies have investigated the acute effects of incorporating RS into meals and the effects of RS on appetite in lean and overweight subjects [19,22,23]. However, the results are inconsistent due to variability in dosages, forms or types of RS, and limitations in the study design. In addition, findings also may be confounded by either BMI categories or gender. Accordingly, the present study aimed to investigate the short-term effects of including a 48-g RS supplement over the course of a day, compared with a placebo in overweight male subjects. The effects of RS on the appetite visual analogue scale (VAS) scores, 24 h food intake records, glycaemia, insulinaemic responses, C-peptide concentrations and gut peptide Glucagon-like peptide (GLP-1) concentrations involved in satiety and actual food intake were examined.

## 2. Materials and Methods

### 2.1. Participants

Healthy, overweight male aged 18–32 years (Table 1) were recruited from the student population at the University of Surrey using flyers and e-mail advertisement. The participants were invited to attend the screening session at the Clinical Research Centre (CRC) to assess their suitability for inclusion. The inclusion criteria were as follows: body mass index (BMI) within 28–37 kg/m²; fasting blood glucose ≤ 6.0 mmol/L; haemoglobin ≥ 13 g/dL, no history of gastrointestinal disease or endocrine disorders; and stable weight for a period of 3 months. The exclusion criteria included previous or current chronic medical conditions, alcohol intake of more than 21 units per week, or any prescription medicines or supplements in the last 6 months. Restrained eaters (restraint score ≥ 4 as identified by the Dutch Eating Behaviour Questionnaire (DEBQ) were also excluded from participating in the study [24]. The participants included in this study had a mean (SEM) score of 2.1 (0.2) on the emotional scale, 2.5 (0.2) on the restraint scale and 3.2 (0.2) on the external eating scale. The study was conducted according to the guidelines of the Declaration of Helsinki, and all procedures involving human subjects were approved by the Ethics Committee of the University of Surrey (EC/2011/80/FHMS). Written informed consent was obtained from all subjects.

### 2.2. Test Products and Supplements

The RS supplement (Hi-Maize® 260 resistant starch comprising 60% RS type 2 and 40% rapidly digestible starch) and the placebo (Amioca, 100% rapidly digestible starch) were manufactured and supplied by Ingredion Incorporated (Bridgewater, NJ, USA). The RS and placebo supplements were mixed into a mousse (Angel Delight, Ashford, UK). To balance the carbohydrate and energy content of the two test products, 32 g of the placebo (16 g in each mousse) was used. In addition, the results from preliminary work by our group [23] showed that 40 g of RS (Hi-Maize) was the maximal amount that could be incorporated into a single portion while still being acceptable in term of taste and texture. As a result, 80 g of Hi-Maize composed of 60% RS type 2 and 40% rapidly digestible starch (RDS)) was therefore divided into two portions over breakfast and lunch to provide 48 g of RS (24 g in each mousse) and 32 g of RDS in a single 24-h period (according to the measurement of The Association of Official Analytical Chemists for total dietary fibre method 991.43) [25]. Therefore, to balance the carbohydrate and energy content of the two test products 32 g of the placebo Amioca (100% RDS), with 0 dietary fibre from RS was divided into two mousse portions over breakfast and lunch to offer 16 g in each mousse in a duration of a 24 h. Accordingly, the glycemic carbohydrate loads for the test product were identical and the only difference was in weight and dietary fibre content.

### 2.3. Study Protocol

This study was a randomized, subject-blinded, balanced crossover study that investigated the acute effect of RS on appetite, glucose, insulin levels and GLP-1 levels. The participants were studied on two separate days, at least one week apart. The participants consumed either 80 g of RS or 32 g of placebo, which was incorporated into breakfast and lunch meals and were assigned randomly. Prior to each study day, all participants were instructed to restrain from strenuous exercise and alcohol consumption in order to reduce within-subject variability. The participants were asked to consume a standard low fibre meal the previous evening, and 24-h intake was recorded. The participants remained at the investigation centre for 7 h with water provided *ad libitum*. On each study day, the participants arrived at the CRC at 8:00 a.m. following an overnight fast of 12 h. Initially, anthropometric measurements (weight, height, waist circumferences, % body fat, BMI) and blood pressure (BP) were recorded. The participants were cannulated, and then, two fasted blood samples were taken: one 15 min prior to breakfast and the other just before breakfast (0 min). Thereafter, baseline appetite was subjectively assessed by 100-mm visual analogue scale (VAS) questionnaires for fullness, hunger, prospective food consumption, desire to eat meal/snack/sweet/savoury/salty/fatty and nausea, as previously described [26].

The participants were then asked to consume a standardised breakfast at time zero (the first study visit) that included 30 g of Rice Krispies cereal (Kellogg’s, Manchester, UK) with 100 g of semi-skimmed milk (Tesco, Welwyn Garden, UK) and the first dose of the supplement (either RS or placebo) mixed into a mousse. Following breakfast, the participants completed a VAS questionnaire regarding the palatability and pleasantness of the mousse. Blood samples were taken, and VAS assessments were performed every 30 min following the breakfast until lunch.

Three hours post-breakfast (180 min), the test lunch, which included a ham or a cheese sandwich, crisps, an orange drink and the second dose of the supplement (either RS or placebo) mixed into a mousse, was provided. Each participant consumed the same sandwich filling on each visit. On the first visit, the participant could eat as much or as little of the provided lunch as they liked, excluding the test product, which needed to be consumed in its entirety. After the lunch, the remaining food was weighed, and the participant was given the same amount of food on the subsequent visit to match the energy and macronutrient intakes, as previously reported [27] and used by our group [23]. Table 2 summarises the energy and macronutrient composition of breakfast and lunch.

The cannula was removed at 420 min, after which the participants were provided with a large pre-weighed homogeneous (*ad libitum*) pasta meal in a quantity exceeding usual portion sizes (1502.2 g). The participants were separated to minimise any effects of social interaction prior to food intake and were told to eat until comfortably satisfied, and then, the remaining food was reweighed. To prevent over-consumption, the participants were told they could take home anything they did not want to eat, as previously conducted by our group [26,28]. Following the *ad libitum* meal, the subjects were free to leave and were asked to complete a food and drink diary for the remainder of the day until 9 the following morning, which in addition to intake at the breakfast, lunch and ad libitum test dinner was used to estimate the overall 24-h intake. All food diaries were analysed using Dietplan 6 version 6.6 (Forestfield Software Ltd., Horsham, UK). The participants recorded their bowel movements on the day of the study and the following day to assess gastrointestinal tolerance of the supplements.

### 2.4. Ad Libitum Pasta Meal

An *ad libitum* homogeneous pasta meal was given to the participants. If completely eaten, the meal provided 9665 kJ of energy, 83.5 g of protein, 335.6 g of carbohydrates, 68.1 g of fat, and 15.9 g of fibre. This meal had a mean weight of 1502.2 (SD: 5.8; SEM: 4.1) g, and the energy density of this meal was 6.2 (SD 0.4, SEM 0.3) kJ/g. The *ad libitum* pasta meals were weighed before and after serving to determine the amount consumed. The energy and macronutrient intakes were then calculated based on the energy density for each individual study day due to minor differences in water weight.

### 2.5. Biochemistry

Blood samples were collected in potassium EDTA tubes for the insulin analysis and in sodium oxalate tubes for the glucose analysis. Potassium EDTA tubes containing 200 kallikrein inhibiting units (KIU) of aprotinin per mL of whole blood (Sigma-Aldrich company Ltd., Gillingham, Dorset, UK) were used for the GLP-1 analysis to prevent enzymatic degradation. All blood samples collected were stored at 4 °C and then centrifuged at 1750× *g* for 10 min. Aliquots of plasma were taken and then stored at −20 °C until analysis. All samples were analysed in one batch at the end of the study to reduce inter-assay variation. Plasma glucose levels were measured enzymatically using a commercially available kit (Instrumentation Laboratory, Warrington, UK) with the ILab650 system (Instrumentation Laboratory); the inter-assay variation was <3%, and the intra-assay variation was <2%. Plasma insulin and C-peptide levels were measured by radioimmunoassay (RIA) with a commercially available kit (Millipore; Missouri). These assays had inter-assay variations of 31.2% and 18.31%, respectively, and intra-assay variations of 12.5% and 9.63%, respectively. Plasma GLP-1 levels were measured by a commercially available enzyme-linked immunosorbent assay (ELISA) kit (Millipore; Missouri) with an inter-assay variation of 17.7% and an intra-assay variation of 4.0%.

### 2.6. Insulin Sensitivity

To evaluate insulin resistance, insulin sensitivity and β-cell secretion capacity, several indexes were estimated based on a web formula (downloadable from http://mmatsuda.diabetes-smc.jp/english.html). The homeostasis model assessment of insulin resistance (HOMA-IR) was determined in the fasting state to assess baseline insulin resistance. The Matsuda Index was estimated postprandially after each meal to assess insulin sensitivity due to meal ingestion [29,30,31,32]. The disposition index, which provides a measurement of β-cell function, was also assessed [33,34]. The insulinogenic index was estimated to evaluate the early phase of insulin secretion [35,36].

### 2.7. Calculations and Statistical Analysis

The area under the curve (AUC) for glucose, insulin, C-peptide and the ratio of C-peptide to insulin was calculated using the trapezoid rule. The AUC for the metabolites and the ratio were determined for up to two hours after each meal (0–120 min and 180–300 min) and for the total until two hours after the test lunch (0–300 min). This ratio was used as a surrogate marker of hepatic insulin clearance [37].

All statistical analyses were conducted using SPSS for Windows 18.0 (SPSS Inc., Chicago, IL, USA), and *p* < 0.05 was considered significant. A repeated measures analysis of variance (ANOVA) was used to assess the effects of the treatment (RS and placebo) and the change over time within subjects as a factor. Paired *t* tests were used to compare groups, and the data were checked for normality using the Kolmogorov-Smirnov test/Shapiro-Wilk’s test (SW/KS). The influence of the RS supplement compared with the placebo on postprandial subjective VAS scores, glucose levels and insulin sensitivity was assessed by two-way repeated measures ANOVA with starch (RS, PL) and time (15 time points) as independent variables and the measurements as a continuous dependent variable. The AUC and insulin sensitivity were compared by paired sample *t* test. Differences in the effect of RS compared with placebo on energy intake at the *ad libitum* meal, energy intake and macronutrient intake over 24-h were assessed by paired sample *t* test. Non-parametric tests were also used due to the small number of participants. However, statistical significance did not differ between tests; thus, parametric tests were reported as the more powerful statistical test. One-way ANOVA was used to compare the differences in the effect of RS compared with the effect of the placebo on the palatability VAS scores. All the results are expressed as the means with their standard errors.

## 3. Results

### 3.1. Quantitative Appetite Assessment

The consumption of 48 g of RS divided between breakfast and lunch resulted in a reduction in the energy intake at the *ad libitum* test meal consumed at dinner compared with that of the placebo (4551 ± 617 kJ versus 5197 ± 561 kJ, respectively; *p* = 0.017). Comparison of the means was made by paired-sample *t*-test Table 3.

The mean daily energy and macronutrient intake over the 24-h period was not significantly different from that of the placebo (*p* = 0.508). As expected, the total dietary fibre intake was significantly higher with RS, as a direct result of the RS supplement (*p* = 0.005) (Table 3).

### 3.2. Qualitative Appetite Assessment

The subjective appetite ratings, measured by VAS scores, revealed no significant differences between the two supplements (RS and placebo) at any time point for hunger (Figure 1), fullness, prospective food consumption, thirst or desire to eat (sweet, salty, savoury or fatty) foods.

### 3.3. Postprandial Metabolites

For the postprandial glucose concentration, RS ingestion resulted in a significant treatment × time interaction (*p* = 0.004), but no significant treatment effect of RS compared with placebo was found (*p* = 0.36) (Figure 2). Glycaemic variability was evaluated and showed no significant differences between meals [38]. No treatment or treatment × time interactions were found regarding the postprandial insulin, fasting or postprandial levels of GLP-1 between treatments. Although there was a trend for higher C-peptide concentrations after RS intake compared with PL (treatment × time, *p* = 0.089), no significant treatment effect of RS consumption compared with PL (treatment, *p* = 0.393) was found. The C-peptide response appears to be different only in the early stages (30–180 min) between breakfast and lunch (Figure 3), However, the C-peptide to insulin ratio was unchanged following RS intake. The AUCs were not significantly different for insulin, while a trend to significance after breakfast AUC (0–120 min) was observed for C-peptide (Table 4) after RS supplement consumption compared with that after placebo consumption (*p* = 0.065). However, the AUCs after the lunch meal and for the total were not significantly different between treatments.

### 3.4. Insulin Sensitivity

No significant difference was found in fasting insulin sensitivity, cell function and insulin resistance as estimated by HOMA. Neither postprandial insulin sensitivity nor early-phase insulin secretion was found to differ between treatments (data not shown).

### 3.5. Palatability and Gastrointestinal Symptoms

The mean VAS scores regarding the pleasantness and palatability of the mousse were not significantly different between the two supplements for breakfast and lunch (pleasant: breakfast (*p* = 0.86) and lunch (*p* = 0.83); palatable: breakfast (*p* = 0.21) and lunch (*p* = 0.27). Analyses were carried out using one-way ANOVA. No adverse gastrointestinal symptoms were reported on either the day of the study or the following day, and both supplements were well tolerated by the participants.

## 4. Discussion

In the present study, we demonstrated that RS acutely and significantly (*p* = 0.017) reduces food intake at an *ad libitum* meal in overweight/obese men compared with placebo. However, this significance effect was lost when the whole 24 h intake was evaluated. Despite the effects found on actual food intake, no significant impact of RS was observed on qualitative feelings of satiety, or hunger.

Our finding that RS reduces overall energy intake at a subsequent test meal agrees with other studies in normal and overweight individuals [22,26,39]. However, the deficit in energy intake in the present study was not maintained over the entire 24-h period, in contrast to research on normal-weight individuals [26]. Despite the observed reduction in food intake, our data indicate that RS did not affect any of the subjective appetite ratings in any of the assessment questions, a finding that agrees with previous data in normal-weight subjects [26]. Research on the effect of RS on energy intake and satiety in humans has thus provided inconsistent results. Subjective satiety is affected by subject variability, indicating that the reduction in food intake may not result from increased satiety and vice versa. Therefore, subjective appetite ratings may not anticipate energy intake, as been observed in this study.

It should be noted, however, that some of the previous studies have shown that the RS supplements were less palatable than the control, which may influence the outcome of studies that examine the effect of RS on food intake [11,21]. In the present study, despite the reduction of food intake, no significant differences were observed in the palatability between RS and PL, indicating that intake was reduced due to a post-ingestive mechanism. Since GLP-1 has been shown to reduce energy intake, improve the sensation of fullness, and reduce the feelings of hunger in lean and obese subjects [24,40], a lack of increase in the GLP-1 concentration in the present study could explain why subjective feelings of satiety did not improve in the present study.

A previous study comparing the effect of consumption of 40 g of RS (native banana starch) with a control (digestible corn starch) in healthy subjects demonstrated a reduction in food intake at the ad libitum test meal 3 h post-ingestion, without any effect on subjective appetite ratings [22]. However, this study did not investigate the impact over 24 h to confirm whether effects were sustained. Also, the preload was mixed with pure water, which may have resulted in differences in the texture between the meals and changed the palatability of the RS. In addition, the RS preload meal given to the subjects was matched by weight of starch and thus varied in energy and available carbohydrate content. Moreover, the study included both lean and overweight subjects of both genders, which may confound their results.

The acute reduction of food intake at the *ad libitum* meal in the present study could be mediated by the fermentation of RS in the colon to produce short chain fatty acids (SCFAs). This has been proposed as a mechanism of dietary fibre to alter appetite and increase satiety [41]. RS consumption has been shown to increase SCFA concentrations in rodent studies [20,42,43]. However, this holds no insight for humans, due to species differences in gut physiology, the amount of RS given to the rodent (higher), and the duration of the experiment in rodents (longer). Therefore, analysing serum SCFA concentrations in humans may provide a more complete understanding and may confirm the effect of RS on fermentation. However, we did not measure the SCFA content in the current study. Therefore, it is not possible to determine the extent of fermentation in the colon due to RS supplementation or to assess whether increased fermentation is the reason for the reduction in energy intake.

Our findings further suggest that RS significantly influences postprandial glucose concentrations (*p* = 0.004). Following RS consumption, there was more ‘stability’ in plasma glucose levels after breakfast, with no evidence of reactive hypoglycaemia. Conversely, plasma glucose levels dropped rapidly to below fasting levels following consumption of the placebo. This RS-induced ‘stability’ in blood glucose levels has been previously noted in both humans and animals [39,43,44,45] and may form part of the explanation for the differences in food intake documented between the two treatment groups.

In this study, both insulin responses and insulin sensitivity were not significantly different between the RS and placebo supplements, a finding that might have been expected in overweight/obese subjects who are insulin resistant [46]. Although insulin secretion maintains blood glucose levels within the normal range, most individuals with insulin resistance, such as the overweight and obese subjects in our study, may have impaired insulin secretion and defective insulin action.

It remains unclear why the RS in the current study influenced energy intake and improved glucose stability but had no effect on either insulin response or insulin sensitivity. To our knowledge, this is the first time that such an observation has been made following the acute administration of RS in obese/overweight individuals. However, the glucostatic theory offers a possible explanation. According to this theory, a rise in blood glucose concentration may stimulate the ’satiety centre’ in the ventromedial hypothalamus (VMH) in the brain, which increases satiety by inhibiting the ‘feeding centre’ and terminating the meal; therefore, reducing energy intake [47,48,49].

Several limitations of the present study need to be addressed. First, although the sample size was small, a within-subject crossover study design was applied to increase the power of the study. Moreover, inclusion criteria were restricted to only overweight males; thus, it is not possible to generalise the findings to overweight females. We may also need to measure the SCFA and gastric emptying to draw a more informed conclusion about the impact of RS on appetite. Since individual gut hormones are not secreted separately after a meal, hormones may be co-secreted with other hormones and may act together to regulate appetite and energy intake [50]. Therefore, the investigation of only one hormone in this study may not provide a complete picture. Further investigations should thus involve the effect of different gut hormones on appetite in addition to GLP-1.

In conclusion, the findings from this study suggest further long-term studies to determine the mechanisms by which RS decreases food intake. Presently, our findings do not support the role of RS in mediating food intake in an overweight/obese group.

## Figures and Tables

**Figure 1 nutrients-10-01993-f001:**
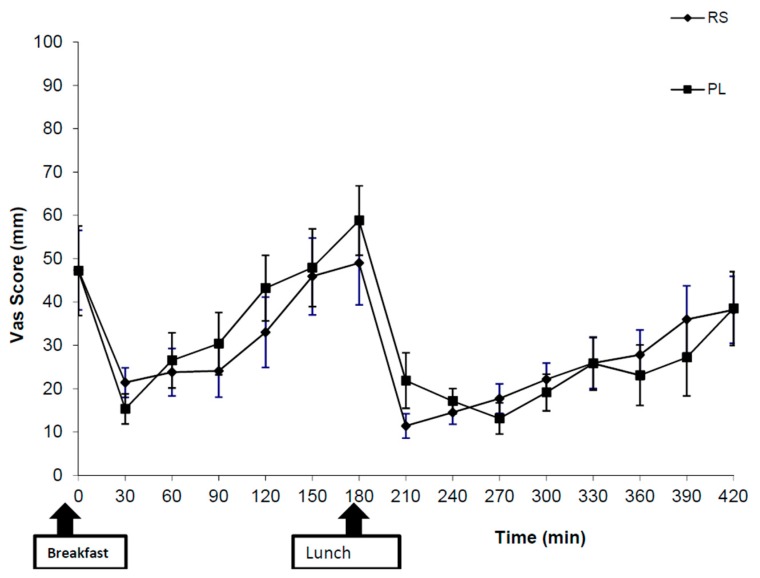
Subjective appetite ratings in response to the question “how hungry do you feel?” after consumption of 48 g of resistant starch (RS) or placebo (PL). The values represent means ± SEM (*n* = 10).

**Figure 2 nutrients-10-01993-f002:**
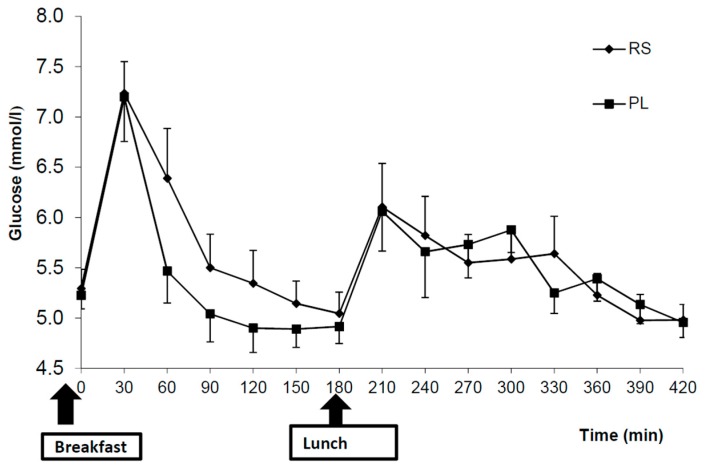
Postprandial plasma glucose concentrations after the consumption of 48 g of resistant starch (RS) compared with a placebo (PL). The values represent means ± SEM (*n* = 10). Comparisons were made with repeated measures ANOVA.

**Figure 3 nutrients-10-01993-f003:**
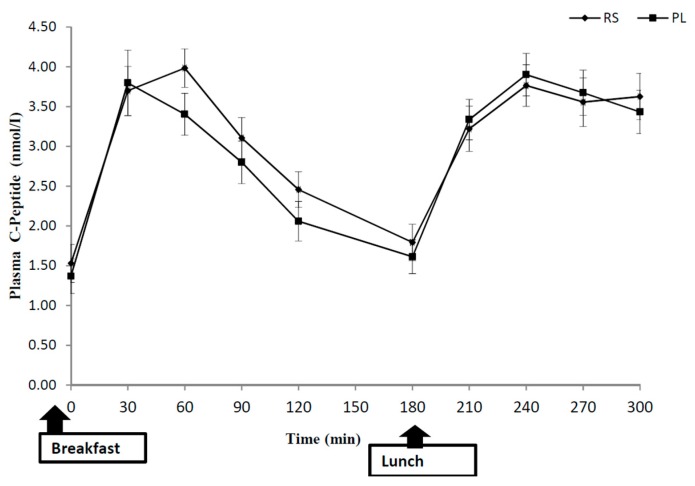
Postprandial plasma C-peptide concentrations after the consumption of 48 g of resistant starch (RS) or placebo (PL). The values represent means ± SEM (*n* = 10). Comparisons were made using repeated measures ANOVA.

**Table 1 nutrients-10-01993-t001:** Baseline participant characteristics in the screening (*n* = 10); the data are shown as the mean values with standard deviation.

	Overall (*n* = 10)		
	Mean	SD	Range
Age (years)	22	3.7	18–31
Height (cm)	176.4	6.5	167–188
Weight (kg)	99.7	12.4	82–119.5
BMI (kgm^2^)	32	2.7	28–37
DEBQ Restrained	2.5	0.70	1.7–3.8
DEBQ Emotional	2.2	0.77	1–4
DEBQ External	3.2	0.71	1.7–4
Fasting Glucose (mmol/L)	4.9	0.65	3.7–5.9
Haemoglobin (g/dL)	16.6	1.52	14.1–19.4

Abbreviations: * BMI: body mass index, DEBQ: Dutch Eating Behaviour Questionnaire.

**Table 2 nutrients-10-01993-t002:** Nutritional composition of the breakfast and lunch meals consumed on both study days. (Mean values with their standard errors (SEM) for ten subjects).

	Breakfast		Lunch	
	Mean	SEM	Mean	SEM
Energy (kcal)	442	1.18	1161	59.1
Protein (g)	9.6	0.05	42.6	0.09
Carbohydrate (g)	83.0	0.33	138	0.34
Fat (g)	8.0	0.05	45.1	4.80
Fibre (g)	0.5 (24.7 *)	0.08 (0.15) *	5.9 (30.1 *)	0.08 (0.15) *

* Resistant starch meal only.

**Table 3 nutrients-10-01993-t003:** Intake following supplementation with 48 g of resistant starch (RS) or matched placebo (PL) supplement (*n* = 10). (**A**) at ad libitum test dinner 420 min postprandially and (**B**) for entire 24 h period postprandially. * (Mean values with their standard errors (SEM) for ten subjects).

	PL		RS		*p*-Value
	Mean	SEM	Mean	SEM	
**(A) Intake of ad libitum Test Dinner**					
Energy (kJ)	5197	561	4551	617	0.017
**(B) 24 h Intake**					
Energy (kJ)	12,553	722.5	12,955	1198.98	0.77
Protein (g)	105.4	8.1	111.7	11.7	0.66
Carbohydrate (g)	419.00	24.0	431.9	36.1	0.75
Sugar (g)	102.3	11.6	97.6	12.3	0.81
Fat (g)	97.61	6.7	99.7	11.89	0.88
SFA (g)	35.32	2.7	41.1	6.1	0.40
Fibre (g)	63.42	1.0	17.2	1.57	0.005

* Comparisons were made by paired sample *t*-test.

**Table 4 nutrients-10-01993-t004:** AUC plasma C-Peptide response, AUC plasma insulin response and C-peptide to insulin ratio for breakfast (0–120 min), lunch (180–300 min) and the total (0–300 min) after the consumption of RS or PL (*n* = 10). Comparisons made by paired sample *t*-test and were not significantly different.

	RS		PL		*p*
	Mean	SEM	Mean	SEM	
*** AUC C-peptide**					
AUC (nmol·L^−1^ 120 min)	383,377	21,302	351,348	0.065	0.065
AUC (nmol·L^−1^ 300 min)	397,467	27,410	3039	22,770	0.58
AUC (nmol·L^−1^ 300 min)	780,844	45,898	754,387	44,383	0.19
*** AUC Insulin**					
AUC 0–120 min (pmol/L)	88,606	10,205	84,766	10,493	0.60
AUC 180–300 min (pmol/L)	96,047	14,930	97,754	15,964	0.86
AUC 0–300 min (pmol/L)	269,460	36,671	228,746	33,646	0.39
*** C-peptide to insulin ratio**					
**B/fast (0–120)**	5.02	0.81	4.68	0.57	0.54
**Lunch (180–300)**	5.13	0.75	5.08	0.72	0.90
**Total (0–300)**	4.93	0.71	4.82	0.63	0.74

* AUC C-peptide (nmol × min·L^−1^); * AUC Insulin (pmol × min·L); * C-peptide to insulin ratio; AUC, area under the curve.

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
