# Peer review of "Acute Effect of Resistant Starch on Food Intake, Appetite and Satiety in Overweight/Obese Males"

_nutrients, 2018, doi:10.3390/nu10121993_

Reviewer 1 Report

This is a well written and interesting paper. It is well referenced and the discussion links the results to earlier research well.

It could do with some minor improvements:

Line 50: Spell out what GLP-1 stands for.

Line 69: Add "deviation" to end of table 1 title.

Study protocol: When reading the study protocol questions arise in the readers mind as to the nature of the placebo and why different weights were used. I wonder if the authors should consider defining the test products and supplements before the study protocol section, rather than after it.

It would be helpful for the actual time of day at which the trial started to be included. It ran for 420 min (7 h) so if subjects started at 8.00 am they would be finished at 3.00 pm. When did they finish their pasta meal? It seems that there was a long time for them to consume more food between finishing the trial and bed time, and this food was not accurately quantified (or was it?). This seems to be a weakness that should be discussed.

Line 247: it is stated that subjects “ --- underreported in their food diaries when self-reporting at home”. Do you need to insert “may have”? If the actual food intake was not accurately (food diaries are not accurate) quantified after the subjects left the clinic, how would you know that they underreported?

Table 3  Please add “resistant starch” before (RS) and (PL) after “placebo” in the title. In the body of the Table it would be helpful to have (A) and (B) in front of the subheadings to link them more clearly to the table heading.ie (A) Intake of ad libitum test dinner. (B) 24 h intake.

Table 4 Although one can work it out from the units, it would be helpful to the reader to have subheadings within the table to make it immediately clear which AUCs refer to which variable, rather than having to work it out from the footnote.

Author Response

We thank the reviewers for their comments and valid suggestions. These have significantly helped improve the manuscript. We have addressed all comments by the reviewers. Please find attached, our detailed response to comments.

Reviewer 2 Report

The authors present a well described trial concerning resistant starch and a myriad of satiety related outcome measurements, which leads to a robust comparison. Sadly the n is small and the trial very short.  

Please find my comments per section below:

Introduction:

Please explain on the basis of literature why this research was done in this BMI and age range

 Methods:

Why 32 gram of this placebo? How was is matched?

What was the mousse that was consumed, and do I understand correctly it was offered separately from the other components of the breakfast or lunch. Was there any instruction or data collection concerning the consumption order of the meal and the mousse?

Line 106: “quantity exceeding usual portion sizes”: please specify how much in grams?

Line 110: “To prevent over-consumption, the participants were told they could take home anything they did not want to eat.” Is this on the basis of literature (if so, which oens?), or anecdotally?

 Please consider moving the test products section up to be before study procedures, to make it more understandable for the reader.

 Line 121: “To balance the carbohydrate and energy content of the two test products, 32 g of the placebo (16 g in each mousse) was used.” This sentence is unclear, and seems to contradict earlier descriptions of the treatments, please clarify this.

 Discussion:

247 “Therefore, the overweight/obese subjects in the present study would appear to fully compensate for a reduced energy intake while at the centre but underreported in their food diaries when self-reporting at home.” I don’t understand this sentence, nor how it relates to the study, please clarify.

 Overall:

low n, not much to be done about this

Ad lib water was offered: was the amount of water consumed recorded? Might the observed effect be due to differences in consumption in water?

Author Response

We thank the reviewers for their comments and valid suggestions. These have significantly helped improve the manuscript. We have addressed all comments by the reviewers. Please attached our detailed response to comments.

Round  2

Reviewer 2 Report

The issues were sufficiently adressed